# PET Imaging in Neuro-Oncology: An Update and Overview of a Rapidly Growing Area

**DOI:** 10.3390/cancers14051103

**Published:** 2022-02-22

**Authors:** Antoine Verger, Aurélie Kas, Jacques Darcourt, Eric Guedj

**Affiliations:** 1Department of Nuclear Medicine & Nancyclotep Imaging Platform, CHRU-Nancy, Université de Lorraine, 54000 Nancy, France; 2IADI, INSERM, UMR 1254, Université de Lorraine, 54000 Nancy, France; 3Department of Nuclear Medicine, Pitié-Salpêtrière Hospital, APHP Sorbonne-Université, 75000 Paris, France; aurelie.kas@aphp.fr; 4Laboratoire d’Imagerie Biomédicale, Sorbonne Université, 75000 Paris, France; 5Department of Nuclear Medicine, Centre Antoine-Lacassagne, Université Côte d’Azur (UCA), 06189 Nice, France; jacques.darcourt@univ-cotedazur.fr; 6Laboratory Transporter in Imaging and Radiotherapy in Oncology (TIRO), UMR E 4320, CEA, UCA, 06189 Nice, France; 7Department of Nuclear Medicine, Assistance Publique-Hôpitaux de Marseille, Aix-Marseille Université, Timone University Hospital, 13000 Marseille, France; eric.guedj@univ-amu.fr; 8CERIMED, Aix-Marseille Université, 13000 Marseille, France; 9Ecole Centrale de Marseille, UMR 7249, Institut Fresnel, Aix-Marseille Université, CNRS, 13000 Marseille, France

**Keywords:** PET, neuro-oncology, FDG, amino acid, somatostatin, glioma, meningioma, primary central nervous system lymphoma, brain metastases, peptide radionuclide therapy

## Abstract

**Simple Summary:**

Positron emission tomography (PET) is a functional imaging technique which plays an increasingly important role in the management of brain tumors. Owing different radiotracers, PET allows to image different metabolic aspects of the brain tumors. This review outlines currently available PET radiotracers and their respective indications in neuro-oncology. It specifically focuses on the investigation of gliomas, meningiomas, primary central nervous system lymphomas as well as brain metastases. Recent advances in the production of PET radiotracers, image analyses and translational applications to peptide radionuclide receptor therapy, which allow to treat brain tumors with radiotracers, are also discussed. The objective of this review is to provide a comprehensive overview of PET imaging’s potential in neuro-oncology as an adjunct to brain magnetic resonance imaging (MRI).

**Abstract:**

PET plays an increasingly important role in the management of brain tumors. This review outlines currently available PET radiotracers and their respective indications. It specifically focuses on ^18^F-FDG, amino acid and somatostatin receptor radiotracers, for imaging gliomas, meningiomas, primary central nervous system lymphomas as well as brain metastases. Recent advances in radiopharmaceuticals, image analyses and translational applications to therapy are also discussed. The objective of this review is to provide a comprehensive overview of PET imaging’s potential in neuro-oncology as an adjunct to brain MRI for all medical professionals implicated in brain tumor diagnosis and care.

## 1. Introduction

With an incidence rate of 12.0/10^5^ per 100,000 person-years, primary central nervous system (CNS) tumors are rare tumors but represent a major public health issue as they are associated with high morbidity and mortality [1]. Among primary brain tumors, glioblastomas are the most commonly occurring malignant brain tumors (14.6% of all primary CNS tumors), whereas meningiomas represent the most commonly detected non-malignant tumors (37.6% of all primary CNS tumors) [2]. Among the malignant brain tumors, with an annual age-adjusted incidence rate of 0.43 per 100,000 persons, primary CNS lymphomas represent the fourth most common cause of malignant brain tumors after glioblastomas, glioma malignant NOS and diffuse astrocytomas [2]. Brain metastases are the most commonly diagnosed CNS tumors and cause significant morbidity and mortality. Brain metastases are estimated to occur as much as 10 times more frequently than primary malignant brain tumors and represent 9–10% of all cancer diagnoses [3]. Lung cancer, breast cancer, and melanomas continue to be the leading primary cancer sites for brain metastases [3].

Positron Emission Tomography (PET) provides insights into the biology of brain tumors that extend beyond MRI and that can be used for differential diagnosis, noninvasive grading, delineating the extent of tumor involvement, for planning surgery and radiotherapy, and for post-treatment monitoring and prognostics. One of the main advantages of PET radiotracers over conventional MRI is that, in addition to image-specific pathophysiological mechanisms, these radiotracers are in most cases independent of disruption of the blood brain barrier (BBB) as opposed to contrast-enhanced MRI sequences [4].

Among the PET radiotracers, 18F-2-fluoro-2-deoxy-D-glucose (^18^F-FDG) is the most widely available and commonly used in clinical nuclear medicine [5]. Tumor cells are characterized by increased glycolytic metabolism mediated through the GLUT-1 transporters, in parallel with increased cell proliferation and a loss of differentiation [6]. The high ^18^F-FDG uptake in surrounding normal brain tissue, specifically due to the GLUT-3 transporters, does, however, limit the use of this radiotracer for imaging specific brain tumors, particularly low-grade gliomas, which may not be visible with ^18^F-FDG [4]. Amino acid PET tracers are characterized by their relatively low uptake in normal grey matter, and can therefore detect gliomas with greater sensitivity than ^18^F-FDG in primary and recurrent tumors which is informative for differentiating recurrent tumors from treatment-induced changes [7]. The three most widely used amino acid PET radiotracers, namely 11C-methyl-methionine (^11^C-MET), O-(2-[18F]-fluoroethyl)-L-tyrosine (^18^F-FET) and 3,4-dihydroxy-6-[18F]-fluoro-L-phenylalanine (^18^F-FDOPA), the latter with a physiological uptake in the basal ganglia, enter cells through the L-amino acid transporters that are overexpressed in gliomas and brain metastases [8,9,10]. However, L-amino acid transporter expression was still present in photopenic gliomas on ^18^F-FET, making it less clear what drives amino acid uptake [11]. The transport and uptake of amino acid PET radiotracers are independent of the blood brain barrier. Because somatostatin receptors (SSTR) are overexpressed in meningioma, particularly SSTR subtype 2 [12,13], radiolabeled SSTR ligands allow to visualize meningiomas. The ^68^Ga-labeled tracers DOTA-D-Phe1-Tyr3-octreotate (DOTATATE) targeting SSTR subtype 2 receptors, and DOTA-Tyr3-octreotide (DOTATOC) targeting SSTR subtype 2 and 5 receptors, are the most used radiotracers in the clinical management of meningiomas. These tracers provide an excellent lesion-to-background contrast, which is due to their low uptake in healthy brain parenchyma [14,15].

Medical professionals, implicated in brain tumor diagnosis and care, typically need to address three fundamental questions that may sometimes not be resolved by conventional MRI: firstly, the initial characterization of the brain lesion, particularly in the context of potentially planning for a future biopsy or for radiotherapy; secondly, the monitoring of treatments to differentiate between recurrence/progression and treatment-related changes; and thirdly, the effectiveness of treatment.

In all instances, the PET information added is relatively easy to access, not only because of tracer availability but also because the data are displayed clearly and interpreted based on multidisciplinary criteria. For example, the visual reading of PET images fused to MRI imaging not only considerably improves the diagnosis but also facilitates the multidisciplinary sharing of information. PET data may thus ultimately modify patient management [16].

An up-to-date summary of the different available PET radiotracers and their current indications for different types of brain tumors in this fast-evolving field would therefore appear informative.

This current review outlines the different indications that PET radiotracers can be used for in the management of gliomas, meningiomas, primary CNS lymphomas, and brain metastases, in current practice, and their advantages over MRI-based applications. Recent advances in radiopharmaceuticals, image analyses and translational applications in peptide receptor radionuclide therapy (PRRT) are also discussed.

## 2. Current Uses of PET Imaging in Brain Tumors

Here, we present the main current indications of the different available PET radiotracers for gliomas, meningiomas, primary CNS lymphomas and brain metastases.

### 2.1. Glioma

The main indications for assessing gliomas using PET imaging, provided in this review, are derived from recommendations of the Response Assessment in Neuro-Oncology working group and the European Association for Neuro-Oncology [7]. 

#### 2.1.1. Initial Characterization

As regards the initial diagnosis, ^18^F-FDG PET allows to identify high-grade gliomas even if its specificity is moderate [6,17]. Delaying PET acquisitions to more than 3 h post injection has been suggested to enhance ^18^F-FDG PET specificity but integrating this type of analysis into the routine practice of nuclear medicine departments remains problematic [18]. The differential diagnosis of other brain tumor entities such as cerebral lymphomas and brain metastases, or of brain infections and inflammatory tissue is also sometimes challenging [19]. Although ^18^F-FDG PET provides a prognostic value in high-grade gliomas [6,20,21] it has only limited value for treatment planning. This is juxtaposed to amino acid radiotracers whose better contrast ratio relative to uptake in healthy brains makes them superior to ^18^F-FDG PET for the differential diagnosis between glioma and non-neoplastic tissue. With 90% sensitivity and specificity, amino acid PET may help differentiate glioma from non-neoplastic tissue, although it is a low priority issue as the overwhelming majority of patients eventually undergo histological evaluation before committing to prolonged cytotoxic therapies [22]. However, a small proportion of gliomas present with no amino acid uptake at all [23], and some inflammation cases are reported with moderate amino acid uptake [24]. 

The prognostic information provided by amino acid radiotracers is significant, as confirmed by the associations between dynamic parameters and histological properties and/or key mutations [25,26,27,28,29] included in the WHO classification [30]. This dynamic analysis, which has been extensively studied in the context of ^18^F-FET PET for many years [31], has been successfully transposed to two other amino acid PET radiotracers: ^11^C-MET [27] and ^18^F-FDOPA [26]. Briefly, dynamic acquisitions begin upon injection of the radiopharmaceutical and last for 30 to 40 min. Two parameters are typically determined from the tumor uptake curves: (i) the time-to-peak (TTP), the interval between the start of acquisition and the maximum uptake value (wash-in), and (ii) the slope, calculated by linear regression of the wash-out phase. A long time-to-peak interval (consistently increasing curve) is classically associated with good tumor prognosis while a short time-to-peak (wash-in followed by a wash-out) is more common in more aggressive tumors. Dynamic parameters of amino acid radiotracers allow to identify gliomas with different outcomes even in populations which include gadolinium-negative gliomas [32]. 

Delineating the extent of glioma involvement, which is essential for planning surgery, is improved by the use of amino acid PET radiotracers, both for low- and high-grade gliomas, since histology-validated series report amino acid uptake in tumor areas extending beyond contrast-enhanced MRI and within nonspecific regions of T2 FLAIR [33,34]. Consistently better prognoses are reported in studies with amino acid PET imaging when surgery is performed with dedicated amino acid PET delineation [35,36,37]. Along this same line, postsurgical amino acid PET volume showed an independent prognostic value for time to recurrence after radio-chemotherapy in a recent prospective glioblastoma trial [37]. Amino acid radiotracers may help to guide the biopsy even in gliomas with a negative contrast-enhanced MRI [38]. Amino acid PET radiotracers may also similarly identify aggressive tumor subregions that may be targeted by radiation therapy, thus limiting marginal or noncentral tumor recurrences [39]. This latter indication does, however, need to be confirmed in larger studies to evaluate its added value to patient outcome. In any case, future amino acid PET imaging studies should implement the Joint EANM/EANO/RANO practice guidelines on standardized analysis of amino acid PET [40] to harmonize data across different centers, facilitate comparability of studies and to build larger databases.

#### 2.1.2. Treatment Monitoring

Early modifications after treatment, namely pseudoprogression within the first 3 months of treatment and radionecrosis after 3 months, currently represent significant challenges in neuroradiology [41]. In this regard and concerning the differential diagnosis between tumor recurrence/progression and treatment-related changes, ^18^F-FDG PET has only a limited added value compared to MRI [42]. Amino acid PET does, however, provide excellent diagnostic performances for the differential diagnosis between tumor progression and pseudoprogression [43] or radionecrosis [44,45,46,47], with sensitivities and specificities of around 80–90%. 

#### 2.1.3. Treatment Effectiveness 

Several clinical trials are currently underway to assess the use of ^18^F-FDG PET for evaluating treatment responses in high-grade gliomas (NCT05212272, NCT02902757). There is currently only limited data which evaluate amino acid PET imaging in the assessment of treatment responses of high grade gliomas [48,49,50,51,52,53]. ^18^F-FET PET was nevertheless able to assess treatment responses after completion of radio-chemotherapy or during adjuvant chemotherapy while conventional MRI was not [50,51]. Similar results were obtained after bevacizumab treatment using ^18^F-FET PET or ^18^F-FDOPA PET [52,53]. However, PET-based radiotherapy in patients with newly diagnosed glioma and PET-based re-irradiation in patients with relapsed glioma have not yet shown any clear evidence of a potential benefit as recently confirmed by the PET/RANO Group [54]. Although data for evaluating treatment responses with amino acid PET in low-grade gliomas are scarce, they are promising as they point to stronger associations between survival and PET responses, compared to responses assessed by MRI [55,56].

### 2.2. Meningioma

The indications reported above are based on recommendations of the Response Assessment in Neuro-Oncology working group [57]. 

#### 2.2.1. Initial Characterization

In meningiomas, SSTR ligands are undoubtedly the radiotracers of choice since meningiomas almost invariably express the SSTR type 2 receptor [58]. Even though significant uptake can be observed in chronic inflammatory tissues, gliomas and brain metastases [15], even higher uptake is observed in meningiomas when using SSTR ligands, with detection sensitivities superior to contrast-enhanced MRI (about 90% vs. 80% for MRI) [14,15]. PET with SSTR ligands is particularly useful in cases of ambiguous, non-contributive MRI and for specific regions which are known to be difficult to appreciate by MRI, such as the base of the skull and bony structures. The rare but aggressive optic nerve sheath meningioma entity almost invariably takes up SSTR ligands [59]. PET with SSTR ligands is thus better suited than contrast-enhanced MRI to delineate tumor margins, which is potentially informative for tumor resection. By the same token, the definition of irradiation volumes prior to radiotherapy could also be improved by performing amino acid PET [60,61,62] or PET with SSTR ligands [63], which invariably and consistently modifies the volumes marked for irradiation compared to those only determined by MRI. This added value is particularly pronounced for tumors located in the previously mentioned regions that are difficult to interpret by MRI, and is also important to potentially spare healthy tissue such as the optic nerve, the chiasm, and the pituitary gland or to reduce irradiation doses to adjacent normal tissue [64]. It should nevertheless be specified that in contrast to the well-established pathological-driven tumor delineation using amino acid PET in glioma [65], the definition of metabolic volume with SSTR ligands in meningioma still remains to be defined. The currently proposed threshold based on an SUV value of 2.3 has been obtained from a small sample size and needs to be replicated before it may be applied to surgical/radiation planning based on PET imaging [14]. In a series of 21 meningioma patients, having undergone amino acid PET and PET with SSTR ligands, every tumor showed high SSTR ligand uptake while two meningiomas remained ^18^F-FET-negative [66]. Tumor grading based on PET imaging is not recommended since it still needs to be evaluated in larger-scale studies. It should nevertheless be noted that SSTR ligand uptake may be reduced in WHO II and is almost absent in WHO III meningiomas because of dedifferentiation of the tumor, which results in the decoupling of SSTR ligand uptake and proliferative activity [67,68]. This does, however, need to be confirmed since the decrease in SSTR2 ligand expression in WHO III meningiomas has not been confirmed in an in vitro study [69]. 

#### 2.2.2. Treatment Monitoring

As far as the differential diagnosis between residual tumor and post-treatment-related changes is concerned, PET with SSTR ligands provides good diagnostic performances to discriminate between meningiomas and other entities, including chronic inflammatory diseases [14]. SSTR PET ligands may therefore be considered to help diagnose recurrent disease. In a series of pretreated transosseous intracranial meningiomas, PET with SSTR ligands showed higher sensitivity (98.5% vs. 53.7%) whilst maintaining high specificity (86.7% vs. 93.3%) when compared to contrast-enhanced MRI [70]. 

#### 2.2.3. Treatment Effectiveness 

The evaluation of meningioma treatment responses using PET is currently still only based on preliminary data [57]. A clinical trial using ^68^GA-DOTATATE to evaluate radiotherapy responses of meningiomas is nevertheless currently in progress (NCT03953131).

### 2.3. Primary CNS Lymphoma

Primary CNS lymphomas are rare brain tumors since they represent only 2% of non-Hodgkin lymphomas and 3 to 6% of primary brain tumors [2]. They are associated with unfavorable prognoses with a median overall survival of 25.3 months [71]. To identify systemic non-CNS lymphomas in patients with lymphomatous brain lesions, the International Primary CNS Lymphoma Collaborative group recommends systemic staging evaluations, which include a whole-body ^18^F-FDG PET [72]. 

#### 2.3.1. Initial Characterization

Even though the same group only recommends brain ^18^F-FDG PET as optional for primary CNS lymphoma, ^18^F-FDG PET may nevertheless still contribute to the diagnosis as it is able to differentiate primary CNS lymphomas from other malignant brain tumors such as glioblastomas and metastases. Indeed, primary CNS lymphoma lesions exhibit great avidity for FDG, with homogeneous uptake and an accuracy of 95% (93–100%) to differentiate between primary CNS lymphomas and glioblastomas [73,74,75]. At initial diagnosis, differential diagnosis between primary CNS lymphomas and infectious lesions can also be achieved with ^18^F-FDG PET in immunocompromised patients [76]. The prognostic value of ^18^F-FDG PET at initial diagnosis has also been recently reported, with the risk of progression increasing by 9% for every 5-unit increase in maximal standard uptake value in a study evaluating Ibrutinib-based regimens [77].

#### 2.3.2. Treatment Monitoring and Effectiveness

Only few studies with small patient numbers have evaluated ^18^F-FDG PET for monitoring treatment [78,79,80] and amino acid PET radiotracers in primary CNS lymphomas [81]. Regarding the treatment response monitoring in primary CNS lymphoma, the three ^18^F-FDG PET studies are concordant in terms of the performance of interim PET during ongoing chemotherapy to predict end-of-treatment complete response [78,79,80]. The high negative predictive values reported for interim ^18^F-FDG PET should encourage the use of PET during ongoing chemotherapy to determine the duration needed for induction therapy, especially when patients do not achieve a complete response on MRI. Regarding progression-free survival prediction specifically, data from the literature are less straightforward: interim ^18^F-FDG PET and interim amino acid PET may predict progression-free survival in primary CNS lymphoma patients [80,81], whereas interim ^18^F-FDG PET may not predict the survival outcome in a larger cohort [79]. No study has to date revealed a significant impact of PET results on overall survival in primary CNS lymphoma. Larger prospective trials in primary CNS lymphoma patients are required to fully assess the true prognostic value of PET imaging on survival outcome. Two clinical trials exploring ^18^F-FDG PET in treatment responses of central nervous system lymphomas (NCT03582254, NCT05083936), the latter also studying the ^18^F-FET PET radiotracer, are currently in progress.

### 2.4. Brain Metastases

The following is based on the Response Assessment in Neuro-Oncology group recommendations [82]. 

#### 2.4.1. Treatment Monitoring

Treatment monitoring of brain metastases is of course highly dependent on controlling the primary extra-CNS disease and should give due consideration to systemic neoplastic disease prior to initiating any treatment changes. Generally, and irrespective of the primary cancer, ^18^F-FDG PET and amino acid PET are recommended for the differential diagnosis of recurrent brain metastases and radiation-induced changes. Radionecrosis after irradiation of brain metastases is common, occurring in about 25% of cases, and contrast-enhanced MRI is not always able to accurately assess the differential diagnosis between radiation-induced changes and recurrent brain metastases [83]. Diagnostic performances of ^18^F-FDG PET for the differential diagnosis of radionecrosis and recurrent brain metastases vary considerably between studies with reported performances, in relatively low numbers of patients [84], ranging from 50% sensitivity/specificity [85] to close to 100% [86]. It appears that like in the case of gliomas, delayed acquisitions may enhance diagnostic performances even though as already discussed this may be difficult to apply in the clinical setting due to camera occupation times [86]. Amino acid PET appears to be superior to ^18^F-FDG PET and perfusion or diffusion weighted MRI for the differential diagnosis of radionecrosis and recurrent brain metastases [85,87]. Diagnostic performances of amino acid PET for this differential diagnosis range from 70–80% for sensitivity and specificity for ^11^C-MET [88] or ^18^F-FDOPA PET [89]. Reported diagnostic performances are even higher when using dynamic ^18^F-FET PET acquisitions in a larger number of patients, with accuracies approaching 90% [90]. Studies with longitudinal follow-up also report higher values for amino acid PET compared to contrast-enhanced MRI for this differential diagnosis (accuracy of 94% for amino acid PET long-term follow-up vs. non-significant for MRI) [91]. In any case, the cost effectiveness of amino acid PET for differentiating recurrent brain metastases and radiation-induced changes has been established in Germany [92].

#### 2.4.2. Initial Characterization

PET imaging of brain metastases is not recommended for other indications, such as initial diagnosis, for which the better spatial resolution of contrast-enhanced MRI remains the preferred imaging technique [93,94]. Furthermore, unlike in glioma, brain metastases are usually well delineated and of small size and amino acid PET does not generally add significant information for biopsy or treatment planning. This may potentially induce a brain metastases selection bias in PET imaging studies, with the inclusion of only centimetric metastases which do not necessarily reflect the overall pool of brain metastases. However, in some clinical situations such as large and heterogeneous tumors, amino acid PET helps identify regions with higher tumor content to better guide biopsies [95]. The differential diagnosis of brain metastases and other malignant brain tumors is nevertheless also challenging when using PET imaging, because ^18^F-FDG PET is highly positive in brain metastases and glioblastomas [96], with brain metastases exhibiting similar levels of expression of the L-amino-acid transporters as gliomas [8]. 

#### 2.4.3. Treatment Effectiveness

PET imaging is currently being evaluated to assess treatment responses and preliminary results, with amino acid PET to monitor immune checkpoint inhibition and targeted therapy appearing promising [97].

A decision flowchart summarizing the main indications of PET imaging radiotracers for the previously mentioned brain tumors and the corresponding levels of 1–3 evidence according to the Oxford Centre for Evidence-based Medicine (OCEBM Levels of Evidence Working Group: “The Oxford 2011 Levels of Evidence”) are provided in Figure 1. Representative PET images of these types of brain tumors are also shown in Figure 2.

## 3. Future Perspectives

PET imaging in neuro-oncology is a rapidly growing area. Here, we present recent advances in radiopharmaceuticals, technological PET systems, image treatment analyses and translational approaches to therapeutic applications.

### 3.1. Advances in Radiopharmaceuticals

Novel physio-pathological brain tumor mechanisms are targeted with new PET radiotracers. Tumor perfusion, angiogenesis and neuroinflammation pathways have attracted particular interest [98]. However, results are very preliminary, and validation studies are required. 

#### 3.1.1. PET Radiotracers for Tumor Perfusion

PET radiotracers for imaging brain tumor perfusion have been used since the 1980s, predominantly including the ^15^O-H_2_O [99] and ^13^N-NH_3_ [100] radiotracers. These radiotracers have the advantage of crossing an intact BBB. They nonetheless have a limited application since their radioactive periods are very short (less than 10 min). In routine practice, they have been surpassed by contrast-enhanced MRI even though the latter is dependent on disruption of the BBB. Another way to image the perfusion of brain tumors is to systematically apply dynamic PET acquisitions which are particularly informative for amino acid PET radiotracers [101]. 

#### 3.1.2. PET Radiotracers for Angiogenesis

Vascularization of brain tumors, in particular high-grade gliomas, can be appreciated with radiotracers targeting angiogenesis, a hallmark of tumor aggressiveness [102]. Arg-Gly-Asp peptide (RGD) PET is an example of this type of angiogenesis-targeting radiotracer which binds αvβ3 integrins, key mediators of the development of new vessels, and which has been evaluated in brain tumors [103,104]. These PET radiotracers are nonetheless dependent on disruption of the BBB but may be useful to monitor antiangiogenic therapies. ^89^Zr-Bevacizumab PET has also been evaluated in pediatric patients to monitor bevacizumab treatment outcomes [105]. Along this same line, PET imaging may also help identify the appropriate individual bevacizumab treatment dose to improve therapeutic efficacy similarly to what has been reported for PET imaging of PD-L1 checkpoint antibodies [106].

#### 3.1.3. PET Radiotracers for Neuroinflammation

Neuroinflammation is another currently widely studied physio-pathological mechanism in brain tumors. One of the neuroinflammation targets that has been investigated in brain tumors is the translocator protein (TSPO), a mitochondrial membrane protein predominantly expressed in mononuclear phagocytes. Its expression in the brain increases in neurodegenerative inflammatory diseases and cancer and is related to the activation of brain auto-immune cells [107]. Even if it is sometimes debated [108], TSPO PET radiotracers seem to be able to cross the BBB [109] as shown in areas exhibiting increased ^18^F-GE-180 both with and without MRI contrast enhancement [110]. TSPO PET, with the historical ^11^C-PK11195 radiotracer [111] or radiotracers with a better signal-to-noise ratio, such as the fluorine-radiolabeled 18F-DPA-714 and 18F-PBR06 tracers which succeeded them [112], has recently been applied to decipher the heterogeneity and dynamics of the tumor microenvironment, specifically in the context of targeted immune therapies and focusing on delineating pro- and anti-glioma immune cell modulation [112]. TSPO PET is able to differentiate potentially aggressive forms of gliomas, graded according to the WHO classification, with a PET-positive rate of 100% among the high-grade gliomas investigated [113]. One of the difficulties that still needs to be overcome is how to compensate for changes in individual patient TSPO affinity due to the presence of single-nucleotide polymorphisms in the gene [114]. 

#### 3.1.4. Other PET Radiotracers

Other PET radiotracers which target specific receptors or factors such as: (i) ^68^Ga-NOTA-NFB and ^68^Ga-Pentixafor which target the expression of C-X-C chemokine receptor type 4 (CXCR4), a cell surface chemokine receptor mediating invasion and metastatic spread [115,116], (ii) ^68^Ga-FAPI-02/04 for imaging inhibitors of fibroblast activation protein (FAP), a protein overexpressed in cancer-associated fibroblast, and (iii) ^68^GA-PSMA for evaluating prostate-specific membrane antigen (PSMA) expression in glioblastoma neovasculature [117], are currently being investigated. They may be useful for translational PRRT but only in high-grade gliomas, as they are all unable to cross an intact BBB.

### 3.2. Advances in PET Systems and Image Analyses

#### 3.2.1. PET Systems

The PET/MRI hybrid system is undoubtedly a technical advance for neuro-oncology imaging [118]. In addition to providing a one-shot PET/MRI procedure, PET/MRI allows to accurately combine information from PET imaging with those of advanced MRI techniques such as perfusion, diffusion, and spectroscopy. This is even more significant when working at the voxel level of radiomics analyses [119] to avoid any misregistration between two imaging techniques acquired with separate imaging systems [120]. Multitude studies have sought to associate and/or compare PET imaging with perfusion MRI [87,121,122], diffusion MRI [123,124,125], and spectroscopy [126,127,128] with PET/MR or to dissociate PET and MRI systems in neuro-oncology. Most of these studies highlight the fact that the information provided by PET imaging and advanced MRI techniques is not redundant but provides complementary information. Available evidence from the literature calls for better performances of amino acid PET over perfusion MRI to detect high-grade tumors, identify tumor recurrence, differentiate recurrence from treatment effects and predict survival, even though both imaging methods are able to discern these indications to some extent [129]. The better performances related to the use of amino acid PET are much more appreciated when compared to diffusion MRI or spectroscopy [129]. These reported differences do, however, underpin the need for prospective studies on larger cohorts and neuropathological validation to further evaluate the differences between these two imaging systems. 

Technological PET advances cannot be separated from improvements in image analyses since the latter are directly dependent on PET system performances. With marked improvements in image signal-to-noise ratios as well as significant potential for further enhancement of spatial resolution, fully digital PET cameras offer high performances for brain PET imaging [130]. The higher detection sensitivity of the latest digital PET systems also tends to offer larger fields of view, and thereby facilitates the development of dynamic analyses [131]. These new PET systems also allow more sophisticated image analyses, namely radiomics, to be developed for neuro-oncology. 

#### 3.2.2. Images Analyses

Radiomics extraction, which corresponds to the voxel-level extraction of morphological, statistical, and textural features, is a challenging and complex process which requires specific steps to be followed to obtain accurate results. Even if it is not currently the case, radiomics measures in neuro-oncology should be prospectively evaluated. In addition, potential confounders such as the higher risk of motion artifacts from patients have to be accounted for, especially for longer scanning times, even though PET/MRI is acquired concurrently. Machine learning models are necessary to interpret results of radiomics features. These advanced image analyses require the development of algorithms based on large datasets, which can sometimes be challenging to obtain for a rare brain tumor and may therefore necessitate data augmentation processes [132]. Several PET radiomics analyses have been performed in neuro-oncology with promising accuracies ranging from 85 to 95% being reported. These include an initial diagnosis study to predict glioma grade with ^18^F-FET [133,134], as well as studies to detect the IDH mutation in glioma with ^18^F-FET [134,135] or ^18^F-FDOPA [136], MGMT promoter methylation with ^18^F-FDG [137] and BRAF mutation status in patients with melanoma brain metastases, although the latter study was only based on a small validation cohort [138]. Although performances with these radiomics analyses using ^18^F-FET PET reach accuracies of around 80%, for glioma recurrences [132,139] and recurrent brain metastases [140], their added value over conventional analyses remains to be defined [141]. It is noteworthy that these radiomics feature extractions can also be extrapolated to dynamic parametric images [28,142] to take into account information provided by both static and dynamic PET acquisitions.

### 3.3. Translation to Peptide Receptor Radionuclide Therapy (PRRT) 

#### 3.3.1. PRRT in Meningiomas

One of the most significant advances in nuclear medicine and neuro-oncology is the development of theranostics, which can currently be used via PET imaging with ^68^Ga-DOTATOC in refractory meningiomas, an entity which cannot be treated by surgery or conventional radiotherapy [143,144]. Indeed, a radiopharmaceutical combining the same molecular vector targeting the SSTR type 2 receptor with a β-emitting radioactive label, yields a high-energy treatment which opens the way to new treatment methods in oncology. This PRRT can be monitored with ^68^Ga-DOTATOC PET imaging. This allows ^68^Ga-DOTATOC PET to serve as a predictive biomarker to monitor outcome and to facilitate individualized treatment optimization in patients with meningiomas. Patients with grade I meningiomas, which are well differentiated, exhibit higher levels of ^68^Ga-DOTATOC PET uptake, and have the best responses to PRRT [67]. ^177^Lu-DOTA-octreotate is a radiopharmaceutical combining a molecular vector targeting the SSTR type 2 receptor with Lutetium 177, a β-emitting radioactive label. ^90^Y-DOTATOC and ^177^Lu-DOTATOC similarly combine a molecular vector targeting the SSTR receptor with a β-emitting radioactive label and can also be used in PRRT of meningiomas. These innovative PRRTs present promising initial results in a meta-analysis of 111 patients with a progression-free survival at 6 months of 61% (95% CI, 50–72) and an overall survival at 12 months of 78% (95% CI, 70–86) [145]. These therapeutic performances should be compared to the actual poor prognosis reported for refractory meningioma with an average rate of progression-free survival at 6 months ranging from 26% (grade II and III) to 29% (grade I). The Response Assessment in Neuro-Oncology group recommends considering as promising any new therapy showing a rate of progression-free survival at 6 months of 30% (grade II and III) to 40% (grade I), which is currently the case for SSTR-targeted PRRTs [143]. PET images before and after ^177^Lu-DOTA-octreotate therapy are presented in Figure 3.

#### 3.3.2. PRRT in Gliomas

PRRT with SSTR receptors has been studied in gliomas [146,147], but SSTR expression is poor, as it is only expressed in 25% of astrocytomas [148]. Pilot studies have evaluated targeting the neurokinin type-1 receptor, which is well expressed in gliomas and whose main ligand is substance P, with β-emitting ^90^Y-DOTAGA-substance P [149] or with α-emitting ^213^Bi-DOTA-substance P radiopharmaceuticals [150], as well as ^177^Lu-PSMA 617 in glioblastoma [151]. It is important to keep in mind that the key to a successful PRRT in glioma relies on the radiotracer of interest crossing the blood brain barrier, at least for targeting tumors without any contrast enhancement. Another challenge to overcome is to ensure sufficient uptake of the associated theranostic radiotracer to guarantee effective PPRT [152].

## 4. Conclusions

To conclude, this review summarizes current recommendations for PET clinical routine imaging of gliomas, meningiomas, primary CNS lymphomas and brain metastases which are supported by extensive literature data and make PET fully mature for routine clinical use in brain tumors. It also discusses recent neuro-oncology advances, specifically new radiopharmaceuticals, developments in image analyses and translational approaches to therapeutic applications, thereby supporting the view that the PET imaging field is only in its infancy.

## Figures and Tables

**Figure 1 cancers-14-01103-f001:**
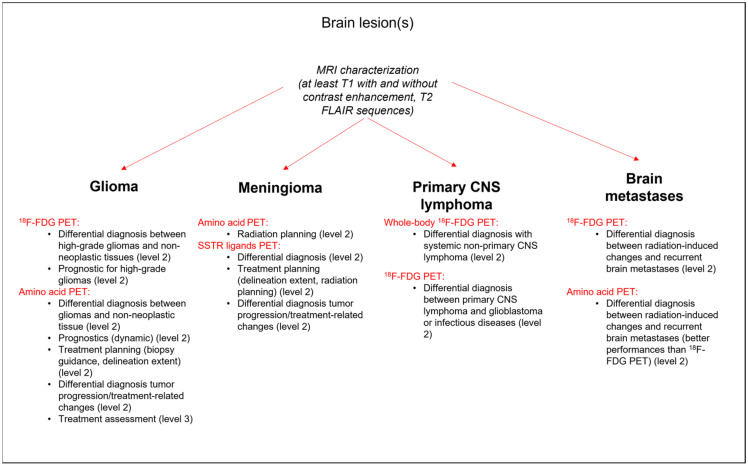
Decision flowchart and main indications for PET imaging radiotracer use according to the type of brain tumor and the corresponding levels of 1–3 evidence according to the Oxford Centre for Evidence-based Medicine (OCEBM Levels of Evidence Working Group: “The Oxford 2011 Levels of Evidence”).

**Figure 2 cancers-14-01103-f002:**
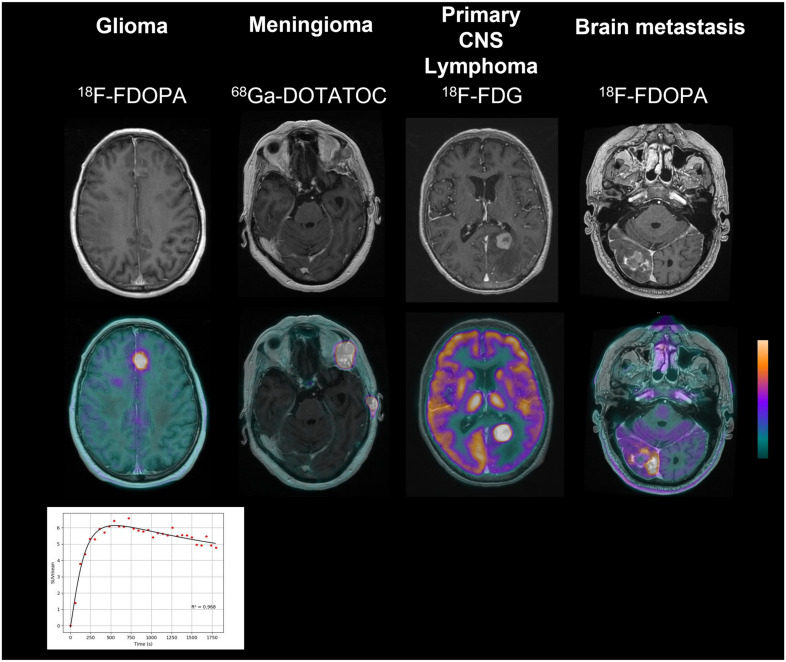
Representative PET imaging of brain tumors. On the left, ^18^F-FDOPA PET at initial diagnosis in a 62-year-old man who presented with a glioblastoma. Note the typical decreasing time–activity curve observed in this lesion with only slight contrast enhancement on MRI. In the middle left, ^68^GA-DOTATOC PET at initial diagnosis in a 70-year-old woman with grade II meningioma. The meningioma is observed in two locations: within the sphenoid bone with an intra-orbital extension on the contrast-enhanced MRI and a transosseous lesion within the temporal bone, not visible on MRI. In the middle right, ^18^F-FDG PET in a 78-year-old woman with a primary central nervous system lymphoma at initial diagnosis. ^18^F-FDG PET exhibits a good contrast ratio between normal brain and the left precuneus lesion. On the right, ^18^F-FDOPA PET identifies the recurrence of a pulmonary adenocarcinoma metastasis in the right occipital brain of a 72-year-old man referred for a differential diagnosis between treatment-related changes and progression related to a lesion, which could not be differentiated on the contrast-enhanced MRI.

**Figure 3 cancers-14-01103-f003:**
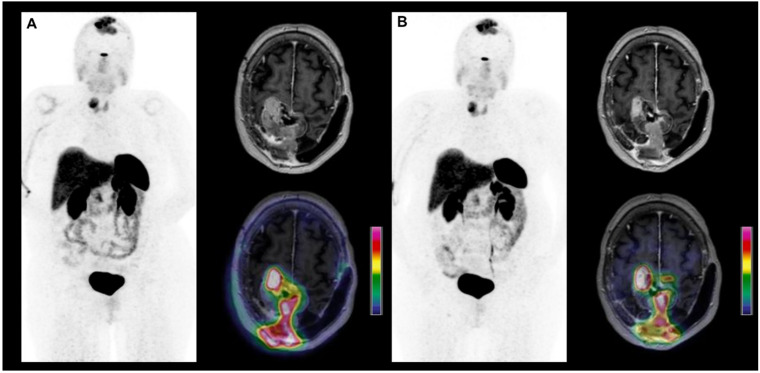
Whole-body maximal intensity projections of ^68^GA-DOTATOC PET and axial slices of contrast-enhanced MRI and ^68^GA-DOTATOC/MRI fused images in a 60-year-old woman patient with a refractory grade I meningioma progressing at a high tumor growth rate, before (**A**) and after 2 courses of ^177^Lu-DOTA-octreotate (**B**). The two PET scans show stable disease.

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
