# Peer review of "PET Imaging in Neuro-Oncology: An Update and Overview of a Rapidly Growing Area"

_cancers, 2022, doi:10.3390/cancers14051103_

Round 1

Reviewer 1 Report

Molecular imaging of brain malignancies is a rapidly evolving study of interest with multiple applications. “Neuro-oncology PET imaging in neuro-oncology: an update and overview of a 2 rapidly growing area” is a review article briefly touching on several of the key areas of concern in neuro-oncology and with additional commentary on potential future advances. The review is well written and researched. There are some minor points that the authors should address.

Minor

Pg 2 line 56. The authors mention GLUT-1 for uptake in tumor cells with FDG uptake. Should mention that GLUT-3 is responsible of high physiologic FDG uptake in the brain parenchyma.

Pg 3 line139. The line, ‘few studies have to date evaluated treatment responses in HGG using AA PET imaging’ is confusing especially given the next line talks about evaluating treatment response. Please clarify the context this sentence refers to.

Page 4 line 159. This statement should to be re-worded. AA PET is able to delineate tumor volume with glioma based on thresholding compared to normal brain parenchyma (1.6 X normal SUV) and has been validated in many studies. A similar process has not been as well established with SSTR PET agents and while this point may be nuanced it is important to address. The article referenced (doi:10.1016/j.ijrobp.2005.12.006) does not actually provide any cut-off for defining tumor volume other an “high uptake” and did not have any pathological correlation nor did it affect management. The referenced article (doi:10.2967/jnumed.114.149120) does correlate uptake with tumor with an SUVmax 2.3. However, this was a small sample size and needs to be better validated before a blanket statement about surgical/radiation planning based on PET should be made.

Pg 4 line 176. Would change ‘therefore recommended to diagnose recurrences” to may consider to help diagnose recurrent disease.

Pg 5 line 177. The patients in reference were untreated meningioma. This is an important point and the authors of this paper have I believe made too strong a conclusion about using SSTR2PET to delineate disease and true recurrent disease.

Author Response

Molecular imaging of brain malignancies is a rapidly evolving study of interest with multiple applications. “Neuro-oncology PET imaging in neuro-oncology: an update and overview of a rapidly growing area” is a review article briefly touching on several of the key areas of concern in neuro-oncology and with additional commentary on potential future advances. The review is well written and researched. There are some minor points that the authors should address.

Response: We thank the Reviewer for his/her insightful comments which we feel have helped improve the quality of this Review.

Minor

Pg 2 line 56. The authors mention GLUT-1 for uptake in tumor cells with FDG uptake. Should mention that GLUT-3 is responsible of high physiologic FDG uptake in the brain parenchyma.

Response: This has been added: ” The high 18F-FDG uptake in surrounding normal brain tissue, specifically due to the GLUT-3 transporters, does however limit the use of this radiotracer for imaging specific brain tumors, particularly low-grade gliomas, which may not be visible with 18F-FDG”.

Pg 3 line 139. The line, ‘few studies have to date evaluated treatment responses in HGG using AA PET imaging’ is confusing especially given the next line talks about evaluating treatment response. Please clarify the context this sentence refers to.

Response: We agree with the Reviewer, and in accordance with the comment of the Editor, we modified this sentence: “There is currently only limited data which evaluates amino-acid PET imaging in the assessment of treatment responses of high grade gliomas”. This sentence now also includes all the references mentioned in the next few lines.

Page 4 line 159. This statement should to be re-worded. AA PET is able to delineate tumor volume with glioma based on thresholding compared to normal brain parenchyma (1.6 X normal SUV) and has been validated in many studies. A similar process has not been as well established with SSTR PET agents and while this point may be nuanced it is important to address. The article referenced (doi:10.1016/j.ijrobp.2005.12.006) does not actually provide any cut-off for defining tumor volume other an “high uptake” and did not have any pathological correlation nor did it affect management. The referenced article (doi:10.2967/jnumed.114.149120) does correlate uptake with tumor with an SUVmax 2.3. However, this was a small sample size and needs to be better validated before a blanket statement about surgical/radiation planning based on PET should be made.

Response: We thank the Reviewer for this important point. We have added the following sentence in the relevant paragraph: “It should nevertheless be specified that in contrast to the well-established pathological-driven tumor delineation using amino-acid PET in glioma [65], the definition of metabolic volume with SSTR ligands in meningioma still remains to be defined. The currently proposed threshold based on an SUV value of 2.3 has been obtained from a small sample size and needs to be replicated before it may be applied to surgical/radiation planning based on PET imaging.“

Pg 4 line 176. Would change ‘therefore recommended to diagnose recurrences” to may consider to help diagnose recurrent disease.

Response: We have implemented the suggestion.

Pg 5 line 177. The patients in reference were untreated meningioma. This is an important point and the authors of this paper have I believe made too strong a conclusion about using SSTR2PET to delineate disease and true recurrent disease.

Response: Thank you for raising this point. We have deleted reference 14, performed in pretreated meningiomas, from the sentence. However, using SSTR2PET to delineate disease and true recurrent disease is consider as level 2 evidence by the PET/RANO group (Galldiks et at Neuro-oncology, 2017). This has now been specified in Figure 1.

Reviewer 2 Report

In this narrative review article by Verger et al, the authors discuss the present and potential future utility of PET across the field of neuro-oncology covering gliomas, meningiomas, PCNSL and metastases. The target audience appears to be neuro-oncologists who may not necessarily have extensive experience with PET. The manuscript’s content is generally high-quality. It is clearly written, although the readability would be greatly enhanced by breaking down subsections into paragraphs (e.g. 2.1 Glioma is a single page paragraph despite various aspects of PET in glioma is covered). 

The two main limitations of this manuscript are the lack of reporting on evidence levels/quality of evidence and the lack of synthesis and identification of basic principles to guide future directions: 

  1. At the very minimum, the supporting level/grade of evidence needs to be reported in the Decision flowchart to inform the readers regarding the robustness of the supporting data
  2. Basic principles and disease-specific priorities in the use of PET in clinical practice are often not delineated or emphasized in the manuscript:
    1. for example, with some rare exceptions, ability to differentiate non-neoplastic from neoplastic diagnoses is a low priority issue as the overwhelming majority of patients eventually  undergo histological evaluation before committing to prolonged cytotoxic therapies.
    2. In glioma I agree with most statements, however, I could not find the data to suggest consistently improved survival by better target delineation, etc. The most recent prospective Biomarker trial for example showed shorter OS with residual MET volume post surgery and the MET volume was included in the radiation planning, but the recurrences still happened mostly locally (Seidlitz et al 10.1158/1078-0432.CCR-20-1775.) Actual survival benefit from using PET in localized therapy planning (if there is data for such) should be highly emphasized.
    3. In meningioma,  it is poorly elaborated on that DOTATATE/DOTATOC/LUTATHERA  that at Grade 3 SSTR2 expression may decrease and thus somatostatin analog uptake decouples from proliferative activity. Statements in the manuscript repeatedly imply a more universal SSTR2 expression in meningiomas. The authors should quote some of the studies looking at SSTR2 expression per meningioma grade (e.g. Sommerauer et al https://doi.org/10.1093/neuonc/now001
      Mahaser et al’s preliminary analysis of DOTATATE-based radiation planning showing reduced radiation dose to adjacent normal tissue using DOTATATE is another benefit to consider adding https://doi.org/10.1093/noajnl/vdab012 
    4. For Primary CNS lymphoma (PCNSL), Line 197-200: the statement regarding treatment monitoring utility of amino acid PET and FDG showing contradictory results (references 67-69) is incorrect. All of these references are concordant about the significant performance of PET to predict PFS. The guiding principle here is again disease specific, because unlike with gliomas, the source of recurrent cells are likely distal (perhaps extra-CNS) thus, the poor correlation between the decrease or loss of tracer uptake (both FDG and MET)  and overall survival might be independent from local response. The low sample size data is actually encouraging to perhaps use PET to determine the duration needed for induction therapy, especially when the patient only achieved CRu and not CR.
    5. For brain metastases the differentiating principles are the existence of extra-CNS disease and the wider availability of targetable tumor alterations. The brain represents a sanctuary site for many tumors, eg Her2+ breast (or some GI tumors), thus making the correct decision whether to change treatment is critical when the patient is already on a therapy that successfully controls the extraCNS disease and there are changes at the site of a previously radiated brain metastasis. 
      The other issue is subcentimeter lesions will alter the sensitivity/specificity or PET and this needs to be further emphasized that the size distribution increases the risk for selection bias in brain metastasis studies.
    6. For “advances in radiopharmaceuticals”, the potential of PET for to enable individualized dose finding is one of the key promises and in Reference 93 this was the purpose rather than monitoring outcomes. Kumar et al 10.1172/JCI122216 is also a good example how peptide-based PET could be used for dose finding in checkpoint inhibition. 
    7. For “Translation to PRRT” section, the key principle is BBB permeability otherwise the target volume could be best treated via resection of the contrast enhancement. These were recently defined as the baseline requirements for novel agents to be considered for Phase 2/3 brain cancer trials by an early phase trial consortium due to the high failure rate of studies. This issue is also reflected by the concerns about theranostic efficacy pf PSMA against GBM (Kirchner et al 2021 10.3389/fonc.2021.774017)
    8. The authors should mention the 2019 Joint EANM/EANO/RANO practice guidelines about standardized analysis of aminoacid PET studies which is key to facilitate the comparability of future studies.

Minor comments (in order of appearance):

Line 66: “the later” should be “the latter”

Line 67: LAT1 expression was still present in photopenic gliomas on FET which makes it less clear what may be the driver of AA uptake https://doi.org/10.1186/s13550-021-00865-9

Line 118: Dynamic parameters should be elaborated (eg. describe Time Activity curves and their interpretation, very briefly so the not PET experienced reader understands

Line 129: more hypothetical tone is more appropriate (e.g. using may instead of can). 

Lines 194-196, ref 65. please mention that this study evaluated Ibrutinib based regimens. These are not (yet) standard of care in PCNSL.

Line 222-224: The cost effectiveness was established in Germany, the statement is overtly generalizing. Unfortunately, such is definitely not established in the USA for example.

Line 228: there is a typo: “planningott” instead of “planning”

Radiomic section: it should be emphasized that radiomics measures should be prospectively evaluated and potential confounders such as higher risk for motion artifacts in patients doing poorly have to be accounted for, especially for longer scanning times even if PET/MRI acquired concurrently. 

Line 344: BRAF detection and radiomics, ref 125. The results are promising but it should be emphasized that the validation cohort was very very small (n=14)

Line 372: the recommendation regarding meaningful PFS-6 in meningioma were made by RANO and note EANO

Lines 376-377: REF135 only included astrocytomas so the statement about low astrocytoma SSTR2 expression would be more accurate. Especially considering TCGA data suggesting SSTR2A expression in oligodendrogliomas with potential favorable prognostic value (Appay et al 10.1186/s40478-018-0594-1)

Author Response

Reviewer 2

In this narrative review article by Verger et al, the authors discuss the present and potential future utility of PET across the field of neuro-oncology covering gliomas, meningiomas, PCNSL and metastases. The target audience appears to be neuro-oncologists who may not necessarily have extensive experience with PET. The manuscript’s content is generally high-quality. It is clearly written, although the readability would be greatly enhanced by breaking down subsections into paragraphs (e.g. 2.1 Glioma is a single page paragraph despite various aspects of PET in glioma is covered).

Response: We have broken down some of the subsections into paragraphs as suggested. We are also very grateful for the Reviewer's extensive and pertinent evaluation which we feel has greatly helped us improve the manuscript.

The two main limitations of this manuscript are the lack of reporting on evidence levels/quality of evidence and the lack of synthesis and identification of basic principles to guide future directions:

At the very minimum, the supporting level/grade of evidence needs to be reported in the Decision flowchart to inform the readers regarding the robustness of the supporting data

Response: We agree with the Reviewer that the level/grade of evidence needs to be reported. We have now included these details in the Decision Flowchart with the different levels defined according to the Oxford Centre for Evidence-based Medicine.

Basic principles and disease-specific priorities in the use of PET in clinical practice are often not delineated or emphasized in the manuscript: for example, with some rare exceptions, ability to differentiate non-neoplastic from neoplastic diagnoses is a low priority issue as the overwhelming majority of patients eventually  undergo histological evaluation before committing to prolonged cytotoxic therapies.

Response: We have now included text delineated in the next few responses to attempt to clarify these issues.  The specific example was addressed by adding the following to glioma subsection 2.1.1: “This is juxtaposed to amino-acid radiotracers whose better contrast ratio relative to uptake in healthy brain, makes them superior to 18F-FDG PET for the differential diagnosis between glioma and non-neoplastic tissue. With 90% sensitivity and specificity, amino-acid PET may help differentiate glioma from non-neoplastic tissue, although it is a low priority issue as the overwhelming majority of patients eventually undergo histological evaluation before committing to prolonged cytotoxic therapies ”.

In glioma I agree with most statements, however, I could not find the data to suggest consistently improved survival by better target delineation, etc. The most recent prospective Biomarker trial for example showed shorter OS with residual MET volume post surgery and the MET volume was included in the radiation planning, but the recurrences still happened mostly locally (Seidlitz et al 10.1158/1078-0432.CCR-20-1775.) Actual survival benefit from using PET in localized therapy planning (if there is data for such) should be highly emphasized.

Response: We have now emphasized this issue in glioma paragraph 2.1.1 as follows and have added the suggested reference. “Consistently better prognoses are reported in studies with amino-acid PET imaging when surgery is performed with dedicated amino-acid PET delineation [35–37]. Along this same line, postsurgical amino acid PET volume showed an independent prognostic value for time to recurrence after radio-chemotherapy in a recent prospective glioblastoma trial.” However, as recently reported by the PET/RANO group, PET-based radiotherapy in patients with newly diagnosed glioma and PET-based re-irradiation in patients with relapsed glioma do not show clear evidence of a potential benefit: “However, PET-based radiotherapy in patients with newly diagnosed glioma and PET-based re-irradiation in patients with relapsed glioma have not yet shown any clear evidence of a potential benefit as recently confirmed by the PET/RANO Group.”

In meningioma,  it is poorly elaborated on that DOTATATE/DOTATOC/LUTATHERA  that at Grade 3 SSTR2 expression may decrease and thus somatostatin analog uptake decouples from proliferative activity. Statements in the manuscript repeatedly imply a more universal SSTR2 expression in meningiomas. The authors should quote some of the studies looking at SSTR2 expression per meningioma grade (e.g. Sommerauer et al https://doi.org/10.1093/neuonc/now001

Response: Thank you for this comment which we did however briefly mention in the previous version of the manuscript by saying that: “It should nevertheless be noted that SSTR ligand uptake may be reduced in WHO II and al-most absent in WHO III meningiomas because of dedifferentiation of the tumor.” We have now expanded on this sentence and added the suggested reference: “It should nevertheless be noted that SSTR ligand uptake may be reduced in WHO II and is almost absent in WHO III meningiomas because of dedifferentiation of the tumor, which results in the decoupling of SSTR ligand uptake and proliferative activity[67,68]. This does however need to be confirmed since the decrease in SSTR2 ligand expression in WHO III meningiomas has not been confirmed in an in vitro study (Graillon et al, J Neurosurg 2017).”

Mahaser et al’s preliminary analysis of DOTATATE-based radiation planning showing reduced radiation dose to adjacent normal tissue using DOTATATE is another benefit to consider adding https://doi.org/10.1093/noajnl/vdab012

Response: Thank you, for this reference, we have now added the following text: “This added value is particularly pronounced for tumors located in the previously mentioned regions that are difficult to interpret by MRI and is also important to potentially spare healthy tissue such as the optic nerve, the chiasm, and the pituitary gland or to reduce irradiation doses to adjacent normal tissue”.

For Primary CNS lymphoma (PCNSL), Line 197-200: the statement regarding treatment monitoring utility of amino acid PET and FDG showing contradictory results (references 67-69) is incorrect. All of these references are concordant about the significant performance of PET to predict PFS. The guiding principle here is again disease specific, because unlike with gliomas, the source of recurrent cells are likely distal (perhaps extra-CNS) thus, the poor correlation between the decrease or loss of tracer uptake (both FDG and MET)  and overall survival might be independent from local response. The low sample size data is actually encouraging to perhaps use PET to determine the duration needed for induction therapy, especially when the patient only achieved CRu and not CR.

Response: Thank you for this  insightful comment. We nuanced our arguments, removed the term "contradictory results" and expanded on the results from the literature for more clarity. “Regarding the treatment response monitoring in primary CNS lymphoma, the three 18F-FDG-PET are concordant in terms of the performance of interim PET during ongoing chemotherapy to predict end-of-treatment complete response [78–80]. The high negative predictive values reported for interim 18F-FDG PET should encourage the use of PET during ongoing chemotherapy to determine the duration needed for induction therapy, especially when patients do not achieve a complete response on MRI. Regarding progression free survival prediction specifically, data from the literature is less straightforward: interim 18F-FDG PET and interim amino-acid PET may predict progression free survival in primary CNS lymphoma patients [80,81], whereas interim 18F-FDG PET may not predict the survival outcome in a larger cohort [79]. No study has to date revealed a significant impact of PET results on overall survival in primary CNS lymphoma. Larger prospective trials in primary CNS lymphoma patients are required to fully assess the true prognostic value of PET imaging on survival outcome“.  Primary CNS lymphomas are however not associated with systemic involvement at diagnosis (otherwise they would be systemic lymphomas with cerebral involvement), except in the case of lymphoma relapses, but these systemic relapses are rare in primary CNS lymphoma according to the literature and have a better prognosis than cerebral relapses (Jahnke et al. J Neurooncol 2006, Provencher et al. Hemta Oncol 2011). In the interest of clarity the manuscript only focuses on primary CNS Lymphomas to avoid any confusion with systemic lymphomas with brain involvement. The term has now been specified thorough the manuscript.

For brain metastases the differentiating principles are the existence of extra-CNS disease and the wider availability of targetable tumor alterations. The brain represents a sanctuary site for many tumors, eg Her2+ breast (or some GI tumors), thus making the correct decision whether to change treatment is critical when the patient is already on a therapy that successfully controls the extraCNS disease and there are changes at the site of a previously radiated brain metastasis.

Response: We agree with that of course, treatment monitoring of brain metastases is highly dependent on controlling extra-CNS disease. Moreover, studies in the literature recommend specific PET radiotracers irrespective of the nature of the primary cancer. We have added the following explanation to that effect: “Treatment monitoring of brain metastases is of course highly dependent on controlling the primary extra-CNS disease and should give due consideration to systemic neoplastic disease prior to initiating any treatment changes. Generally, and irrespective of the primary cancer, 18F-FDG PET and amino-acid PET are recommended for the differential diagnosis of recurrent brain metastases and radiation induced changes.”

The other issue is subcentimeter lesions will alter the sensitivity/specificity or PET and this needs to be further emphasized that the size distribution increases the risk for selection bias in brain metastasis studies.

Response: We thank the Reviewer for this comment. We have now added that: “This may potentially induce a brain metastases selection bias in PET imaging studies, with the inclusion of only centimetric metastases which do not necessarily reflect the overall pool of brain metastases.”

For “advances in radiopharmaceuticals”, the potential of PET for to enable individualized dose finding is one of the key promises and in Reference 93 this was the purpose rather than monitoring outcomes. Kumar et al 10.1172/JCI122216 is also a good example how peptide-based PET could be used for dose finding in checkpoint inhibition.

Response: We thank the reviewer for this interesting comment. We have now added the following sentence with the suggested reference: “Along this same line, PET imaging may also help identify the appropriate individual bevacizumab treatment dose to improve therapeutic efficacy similarly to what has been reported for PET imaging of PD-L1 checkpoint antibodies [106].”

For “Translation to PRRT” section, the key principle is BBB permeability otherwise the target volume could be best treated via resection of the contrast enhancement. These were recently defined as the baseline requirements for novel agents to be considered for Phase 2/3 brain cancer trials by an early phase trial consortium due to the high failure rate of studies. This issue is also reflected by the concerns about theranostic efficacy pf PSMA against GBM (Kirchner et al 2021 10.3389/fonc.2021.774017)

Response: Thank you for this important comment. We have added the following along with the suggested reference: “It is important to keep in mind that the key to a successful PRRT in glioma relies on the radiotracer of interest crossing the blood-brain-barrier, at least for targeting tumors without any contrast enhancement. Another challenge to overcome is to ensure sufficient uptake of the associated theranostic radiotracer to guarantee effective PPRT [152]”

The authors should mention the 2019 Joint EANM/EANO/RANO practice guidelines about standardized analysis of aminoacid PET studies which is key to facilitate the comparability of future studies.

Response: This has now been added: “In any case, future amino-acid PET imaging studies should implement the Joint EANM/EANO/RANO practice guidelines on standardized analysis of amino-acid PET [40] to harmonize data across different centers, facilitate comparability of studies and to build larger databases.”

Minor comments (in order of appearance):

Response: All the changes suggested in the minor comments have been implemented as suggested.

Line 66: “the later” should be “the latter”

Line 67: LAT1 expression was still present in photopenic gliomas on FET which makes it less clear what may be the driver of AA uptake https://doi.org/10.1186/s13550-021-00865-9

Response: We added the following sentence and included the reference: “However, L-amino acid transporter expression was still present in photopenic gliomas on 18F-FET, making it less clear what may drive amino-acid uptake [11].”

Line 118: Dynamic parameters should be elaborated (eg. describe Time Activity curves and their interpretation, very briefly so the not PET experienced reader understands

Response: We added: “Briefly, dynamic acquisitions begin upon injection of the radiopharmaceutical and last for 30 to 40 minutes. Two parameters are typically determined from the tumor uptake curves: (i) the time-to-peak (TTP), the interval between the start of acquisition and the maximum uptake value (Wash-in), (ii) the slope, calculated by linear regression of the wash-out phase. A long time-to-peak interval (consistently increasing curve) is classically associated with good tumor prognosis while a short time-to-peak (wash-in followed by a wash-out) is more common in more aggressive tumors.”

Line 129: more hypothetical tone is more appropriate (e.g. using may instead of can).

Lines 194-196, ref 65. please mention that this study evaluated Ibrutinib based regimens. These are not (yet) standard of care in PCNSL.

Response: This has now been mentioned.

Line 222-224: The cost effectiveness was established in Germany, the statement is overtly generalizing. Unfortunately, such is definitely not established in the USA for example.

Response: We softened the statement: “In any case, the cost-effectiveness of amino acid PET for differentiating recurrent brain metastases and radiation-induced changes has been established in Germany.”

Line 228: there is a typo: “planningott” instead of “planning”

Radiomic section: it should be emphasized that radiomics measures should be prospectively evaluated and potential confounders such as higher risk for motion artifacts in patients doing poorly have to be accounted for, especially for longer scanning times even if PET/MRI acquired concurrently.

Response: We have added the following sentence: “Even if it is not currently the case, radiomics measures in neuro-oncology should be prospectively evaluated. In addition, potential confounders such as the higher risk of motion artifacts from patients have to be accounted for, especially for longer scanning times even though PET/MRI is acquired concurrently.”

Line 344: BRAF detection and radiomics, ref 125. The results are promising but it should be emphasized that the validation cohort was very very small (n=14)

Response: This has been added.

Line 372: the recommendation regarding meaningful PFS-6 in meningioma were made by RANO and note EANO

Response: This has been corrected.

Lines 376-377: REF135 only included astrocytomas so the statement about low astrocytoma SSTR2 expression would be more accurate. Especially considering TCGA data suggesting SSTR2A expression in oligodendrogliomas with potential favorable prognostic value (Appay et al 10.1186/s40478-018-0594-1)

Response: Thank you, this has now been specified.

Reviewer 3 Report

Describe if there are any trials in the world underway that incorporate PET as a test to evaluate response.

Author Response

Describe if there are any trials in the world underway that incorporate PET as a test to evaluate response.

Response: We searched for trials currently recruiting to assess brain tumor treatment responses with PET imaging in the Clinical Trials database (https://clinicaltrials.gov/) using the key words: “glioma; PET”, “glioma; dopa “, “glioma; FET”, “glioma; methionine”, “central nervous system lymphoma; PET”, “meningioma; PET” and “brain metastases; PET”. We only found two 18F-FDG PET studies for evaluating treatment responses in high-grade gliomas (NCT05212272, NCT02902757), two studies exploring 18F-FDG PET in the treatment responses of central nervous system lymphomas (NCT03582254, NCT05083936), the latter also included the 18F-FET PET radiotracer, and one study investigating 68GA-DOTATATE to evaluate the meningioma radiotherapy responses (NCT03953131). All these currently active trials have been added to the respective paragraphs of the manuscript sections.

Round 2

Reviewer 2 Report

Concerns are all addressed.

Reviewer 3 Report

I think it would be appropriate to clarify what method they have used to select the bibliography